# Intracellular polarization of RNAs and proteins in the human small intestinal epithelium

**Roy Novoselsky**[1], **Yotam Harnik**[1], **Oran Yakubovsky**[1,2,3], **Corine Katina**[4], **Yishai Levin**[4], **Keren Bahar Halpern**[1], **Niv Pencovich**[2,3], **Ido Nachmany**[2,3], **Shalev Itzkovitz**[1] *

**1** Department of Molecular Cell Biology, Weizmann Institute of Science, Rehovot, Israel, **2** Department of General Surgery and Transplantation, Sheba Medical Center, Ramat Gan, Israel, **3** Faculty of Medicine, Tel Aviv University, Tel Aviv, Israel, **4** The De Botton Protein Profiling, The Nancy and Stephen Grand Israel and Health Sciences National Center for Personalized Medicine, Weizmann Institute of Science, Rehovot, Israel

* shalev.itzkovitz@weizmann.ac.il

**Data Availability Statement:** The raw data generated in this study, along with the code used to process the data and generate all figures, are available for download from Zenodo (https://doi.

## Abstract

The intestinal epithelium is a polarized monolayer of cells, with an apical side facing the lumen and a basal side facing the blood stream. In mice, both proteins and mRNAs have been shown to exhibit global basal-apical polarization; however, polarization in the human intestine has not been systematically explored. Here, we employed laser-capture microdissection to isolate apical and basal epithelial segments from intestinal tissues of 8 individuals and performed RNA sequencing and mass-spectrometry proteomics. We find a substantial polarization of mRNA molecules that largely overlaps polarization patterns observed in mice. This mRNA polarization remains consistent across different zones of the intestinal villi and is generally correlated with the polarization of proteins. Our protein analysis exposes streamlined intracellular nutrient transport and processing and reveals that mitochondria and ribosomes are less polarized in humans compared to mice. Our study provides a resource for understanding human intestinal epithelial biology.

## Introduction

The small intestinal epithelium is a monolayer of simple columnar epithelial cells that are tasked with nutrient absorption and defense against microbiota. These cells operate on villi, repeating structures that protrude into the lumen [1]. Intestinal epithelial cells consist of an apical side, which is lined with microvilli for nutrient absorption, and a basal side that interfaces with the stroma and extracellular matrix [2]. Identifying molecular differences between the apical and basal epithelial layers could provide important insights into cellular functions. A recent study comprehensively mapped the apical/basal bias of the entire transcriptome and proteome of mouse intestinal epithelium at a high-throughput using laser capture microdissection (LCM). Besides extensive polarization of mRNAs and proteins, the study revealed that mitochondria are basally polarized, and ribosomes are apically polarized, suggesting that mRNA polarization in mice might serve to regulate translation rates [3]. Polarization patterns in the human intestinal epithelium have not been systematically explored.

Here, we used LCM to assemble the ratios between apical and basal mRNAs and proteins—henceforth termed "apicome," of the human and mouse small intestines. We collected the

org/10.5281/zenodo.13919688). The code is also available at https://github.com/roynov01/Apicome.

**Funding:** S.I. is supported by the Moross Integrated Cancer Center, the Helen and Martin Kimmel Award for Innovative Investigation, the Yad Abraham Research Center for Cancer Diagnostics and Therapy, the Israel Science Foundation grants no. 908/21 and no. 3663/21, the European Research Council (ERC) under the European Union's Horizon 2020 research and innovation programme grant no. 768956, a Weizmann-Sheba joint research grant and a research grant from the Ministry of Innovation, Science and Technology, Israel. The funders did not play any role in the study design, data collection and analysis, decision to publish, or preparation of the manuscript.

**Competing interests:** The authors have declared that no competing interests exist.

**Abbreviations:** DDA, data dependent acquisition; DGE, differential gene expression; ECM, extra cellular matrix; FA, formaldehyde; GSEA, gene set enrichment analysis; HDL, high density lipoprotein; iBAQ, intensity-based absolute quantification; IPMN, intraductal papillary mucinous neoplasm; LCM, laser capture microdissection; LPS, lipopolysaccharides; PBS, phosphate-buffered saline; PCA, principal component analysis; PDAC, pancreatic ductal adenocarcinoma; PNLIP, pancreatic lipase; SLC, solute carrier protein; smFISH, single-molecule fluorescence in situ hybridization; UMI, unique molecular identifier.

apical and basal sides at spatially matching zones along the villi axes from both species and found that the human epithelial cells exhibit profound mRNA and protein polarization. Our data reveals important aspects of human intestinal epithelial cells in an in vivo context.

## Results

### Reconstructing the apicome of the human and mouse intestinal epithelium

To map the mRNA and protein apicome in the human and mouse intestinal epithelium, we used LCM, followed by bulk RNA sequencing (RNA-seq) and mass spectrometry proteomics (Fig 1A). We collected the proximal jejunums from 8 patients who have undergone pancreaticoduodenectomy (Whipple surgery) due to pancreatic pathologies (Fig 1B and S1 Table). In these procedures, the proximal small intestine is removed together with the head of the pancreas that harbors a tumor. In these patients, the intestine is considered disease-free [4]. Additionally, we collected the jejunum from 5 healthy adult mice. To identify potential differences in the apicome along the villus axis, we collected samples from the apical and basal compartments, in both the bases and tips of villi (Fig 1A and 1C). Epithelial segments were carefully dissected to avoid inclusion of stroma. Since the basal compartment in human is significantly smaller compared to the apical compartment (width of 15.4 μm versus 6.6 μm, S1A Fig and S2 Table), this resulted in generally lower mRNA yields in the basal compartments (S1B Fig). We therefore normalized the data in each compartment to enable comparison of mRNA and protein concentrations (Methods).

### Human intestinal epithelium exhibits mRNA polarization

To map the mRNA polarization in the apical and basal compartments, we performed RNA-seq on the LCM samples (S3 Table). The samples clustered by the apical/basal origins and villi zones rather than by patients (S1C Fig). For each gene, we calculated the ratio between apical and basal transcript normalized abundances (Methods). The mouse log2 (apical/basal) significantly correlated with those shown by Moor and colleagues [3] (S1D Fig).

We measured 8,103 genes above mean normalized expression of $10^{-5}$, 632 of which exhibited significant mRNA apical-basal polarization (sign-rank test q-value <0.25, Fig 2A). These

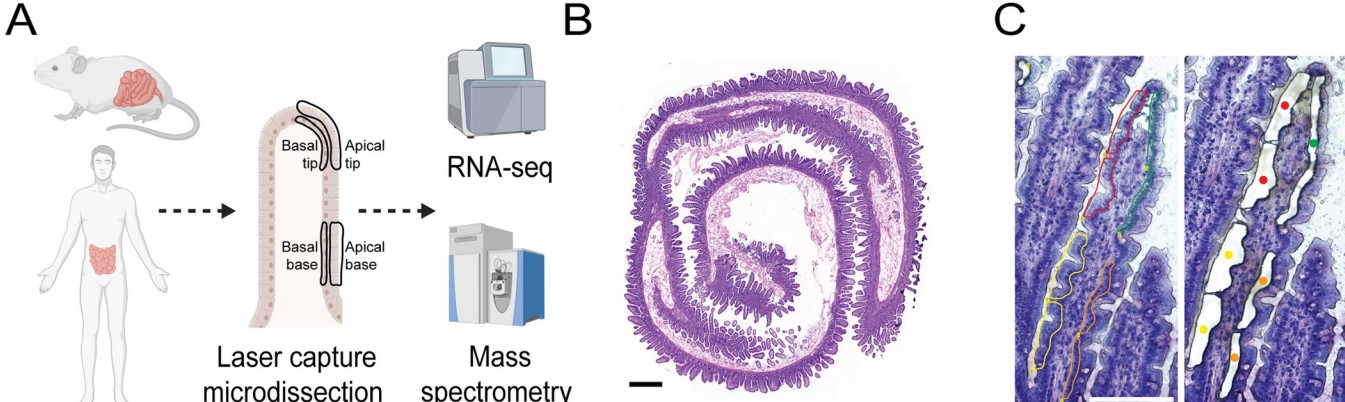

**Fig 1. Reconstructing the apicome of the human and mouse intestinal epithelium. (A)** Diagram of experimental setup. LCM was used on human and mouse small intestine tissues. Apical and basal sides of the epithelial cells were collected from the bases and tips of villi, followed by high-throughput RNA-seq and mass spectrometry proteomics. Created with BioRender.com. **(B)** Hematoxylin and eosin staining of human jejunum, obtained during Whipple surgery. Scale bar, 1 mm. **(C)** Human villus colored with HistoGene staining solution before (left) and after (right) LCM. Colors indicate collection samples—apical tip (red), apical base (yellow), basal tip (green), and basal base (orange). Scale bar, 150 μm. LCM, laser capture microdissection.

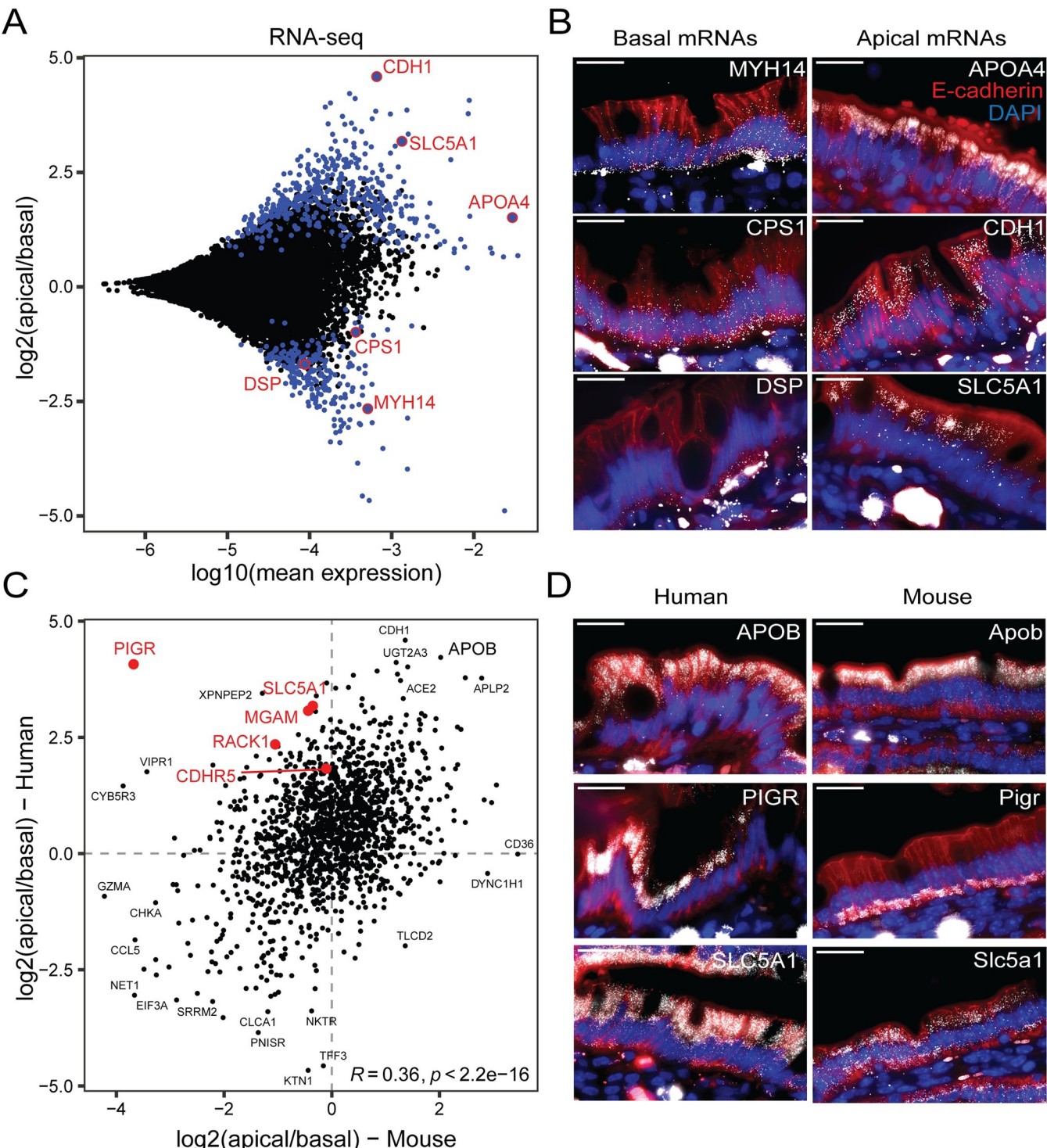

**Fig 2. Human intestinal epithelium exhibits mRNA polarization. (A)** MA-plot of RNA-seq of human samples collected by LCM from jejunums of 8 patients. Significantly polarized genes are in blue (signed rank test, q-value <0.25). Selected genes are circled in red. **(B)** smFISH staining of selected basal (left) and apical (right) genes in human jejunum. Red staining is anti E-cadherin antibody which stains epithelial basolateral membranes, blue staining is DAPI which stains DNA. Scale bar is 20 μm. **(C)** Spearman correlation between the mRNA log2(apical/basal) of human and mouse genes. Genes that significantly change their apicome between mouse and human (Methods) are annotated in red. Other selected genes are annotated in black. Data is available in S3 Table and S1 Data. **(D)** smFISH staining of selected genes in mouse and human jejunums. Red staining is anti E-cadherin antibody which stains epithelial basolateral membranes, blue staining is DAPI which stains DNA. Scale bar is 20 μm. Data is available in S3 Table and in S1 Data. LCM, laser capture microdissection; smFISH, single-molecule fluorescence in situ hybridization.

included the basally polarized genes *MYH14*, *DSP*, and *CPS1* and the apically polarized genes *CDH1*, *SLC5A1*, and *APOA4*, which we validated using single-molecule fluorescence in situ hybridization (smFISH, Figs 2B, S1E, and S1F and S2 Table).

The apical bias of mRNAs had a negative correlation with protein lengths, and positive correlation with mRNA expression levels and stabilities, as well as with translation rates, protein abundances, and stabilities (S2A Fig). Therefore, apical mRNAs seem to be shorter, more stable, and more efficiently translated. Additionally, the apical genes were enriched in metabolic functions such as oxidative phosphorylation, glycolysis, and fatty acid metabolism, while the basal genes were enriched in mitotic spindle pathway genes (S2B Fig and S3 Table). None of the highly expressed genes changed their polarization between the base and tip of the villi in either human or mouse (S2C Fig, no genes had q-value lower than 0.5). Our data therefore exposes mRNA polarization in the human intestinal epithelium that seems to be invariable in distinct villi zones.

Next, we compared the inter-species changes of the mRNA polarization between humans and mice. Apical bias was significantly correlated between the 2 species (R = 0.36, $p = 2.2 \times 10^{-16}$, Fig 2C). Only 5 genes changed their polarization between the 2 species and were also significantly polarized in at least one of them—*PIGR*, *SLC5A1*, *RACK1*, *MGAM*, *CDHR5* (Fig 2C). We validated 2 of these genes with smFISH–*PIGR* was significantly apical in humans yet significantly basal in mice, whereas *SLC5A1* was significantly apical in humans yet non-polarized in mice. We also validated APOB, which shows apical polarization in both species (Fig 2D). As inflammation has been shown to up-regulate Pigr expression via cytokines [5], we checked whether the polarization of Pigr mRNA changes in germ-free mice, antibiotic-treated mice, and lipopolysaccharides (LPS)-injected mice. We found that the polarization remained basal in all conditions (S2D Fig).

## Polarization of mRNA and proteins largely overlap in human

To explore the human protein apicome, we performed LCM followed by mass spectrometry proteomics, identifying 1,582 proteins (S4 Table). We found a significant correlation between the zonated changes in mRNAs and proteins along the villus axis (S3A Fig, Methods). This positive correlation is in line with recent analysis of whole epithelial segments [4]. Among the measured proteins, 288 (approximately 18%) demonstrated significant apical/basal bias (DESeq2, q-value <0.25, Fig 3A). These included the apical proteins encoded by the genes *ANPEP*, *ACE2*, and *SLC5A1* and the basal proteins encoded by *CASK*, *ITGA6*, and *GPA33* (Fig 3A and 3B). Proteins previously shown to be enriched in the mouse epithelial brush border [6] were strongly apical in our measurements (S3B Fig).

We next investigated if an mRNA and its respective protein would localize to the same intracellular compartment. We found a weak, yet significant positive correlation between the apical bias of mRNAs and their matching proteins (Spearman's R = 0.13, $p = 7.8 \times 10^{-6}$, Fig 3C), with the majority of genes exhibiting co-localization of proteins and mRNAs (61%, S3C Fig). An example of a concordant gene was SLC5A1, which showed both apical mRNA and protein polarization (Fig 3C and 3D). Notably, some genes also exhibited discordant mRNA-protein polarization, such as CDH1, with apically polarized mRNAs and basolateral polarized E-Cadherin proteins, and Myosin heavy chain 14 (MYH14), which exhibits the opposite mRNA/protein polarization pattern (Fig 3C and 3D). Although our LCM was carefully performed to focus on the epithelial layer (Fig 1C), the basal compartment included some stromal and immune mRNAs and proteins, which could affect protein-mRNA correlations. We also computed the mRNA-protein correlation using enterocyte-specific genes (Methods, S3 Table) and found a slightly higher positive correlation between the apical bias of mRNAs

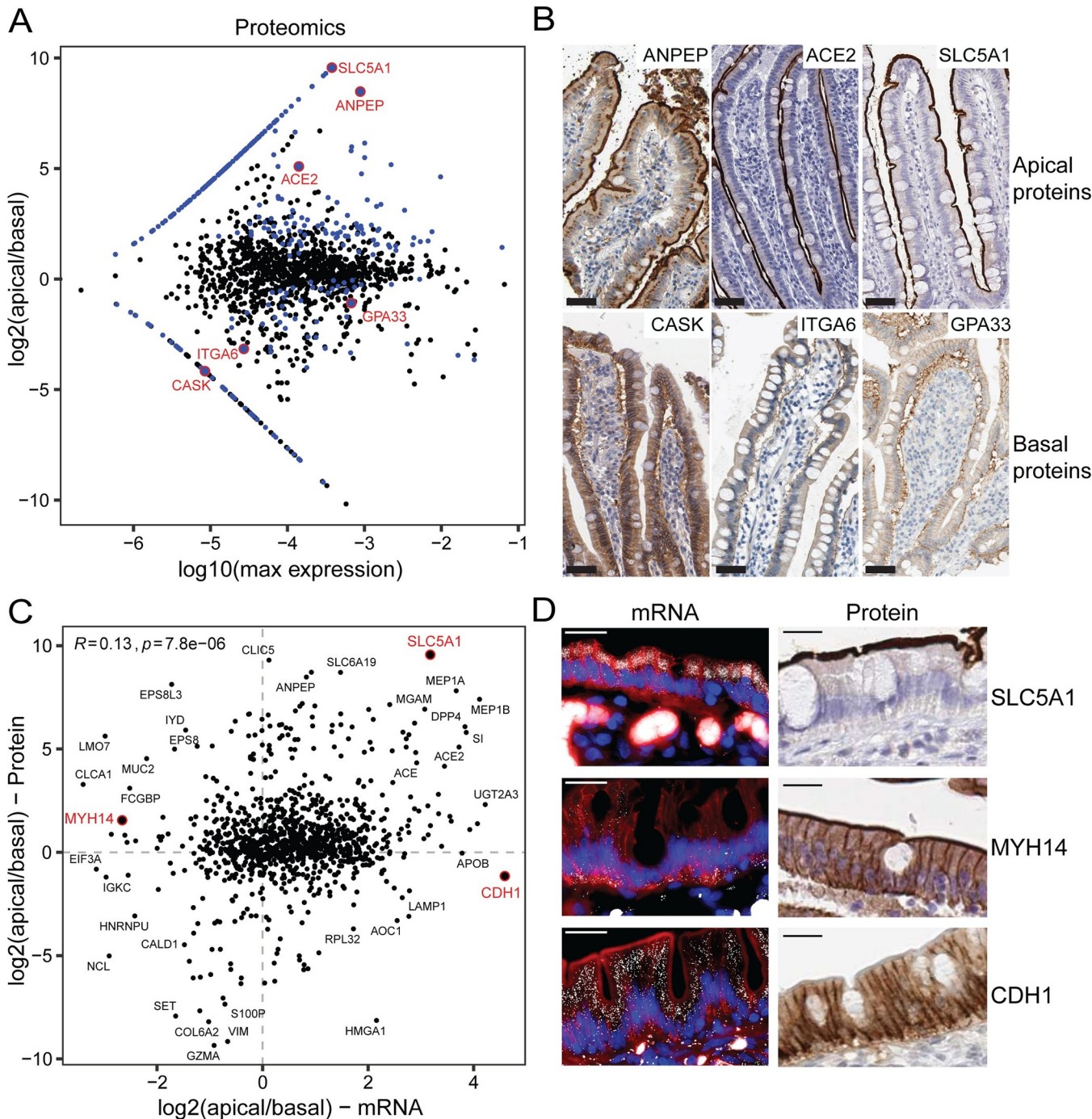

**Fig 3. Human intestinal epithelium exhibits protein polarization. (A)** MA-plot of proteomics of human samples collected by LCM, from jejunums of 4 patients, followed by mass spectrometry. Significantly polarized genes are in blue (q-value <0.25). Selected genes are circled in red. **(B)** IHC staining of selected basal (top) and apical (bottom) proteins from the human protein atlas [7] of healthy duodenum or small intestines. Scale bar is 50 μm. Direct URLs: ANPEP, ACE2, SLC5A1, CASK, ITGA6, GPA33. **(C)** Log2(apical/basal) of mRNAs and their corresponding proteins. R is Spearman correlation, selected genes are annotated in black, and red genes are shown in D. RNA-seq and proteomics data are available in S3 and S4 Tables, respectively. **(D)** (Left) smFISH images of human jejunum, of selected genes, with DAPI in blue and CDH1 protein in red. (Right) IHC images from the human protein atlas [7] of healthy duodenum or small intestines, direct URLs: CDH1, SLC5A1, MYH14. Scale bar is 20 μm. Data is available in S1 Data. IHC, immunohistochemistry; LCM, laser capture microdissection; smFISH, single-molecule fluorescence in situ hybridization.

and their matching proteins (Spearman's R = 0.32, $p = 3.7 \times 10^{-3}$, S3D Fig). This suggests that some proteins and mRNAs in human enterocytes co-localize in their apical/basal cellular compartments.

## Streamlined polarization of protein groups in the human intestinal epithelium

We next focused the analysis on polarization of selected protein groups (Figs 4A, 4B, and S4). As expected, histones showed higher abundances in the basal side, since the nuclei of enterocytes are basally shifted. Extra cellular matrix (ECM) proteins, as drawn from the Matrisome [8] and integrins, were also basal, whereas mucus components, including membrane-associated and secretory mucins [9] were highly apical (Figs 4A, 4B, and S4). Various digestive enzymes, xenobiotic metabolism proteins, together with sulfotransferases were apically polarized. Solute carrier proteins (SLC), which transport various nutrients from the lumen, and particularly amino-acid transporters, showed apical polarization (Figs 4A, 4B, and S4). As reported previously [10], the proteins SGLT1 (*SLC5A1*) and GLUT5 (*SLC2A5*) which transport glucose and fructose, respectively, from the lumen into the enterocytes, were apically localized, whereas GLUT2 (*SLC2A2*), which transports the monosaccharides from the enterocytes to the stroma was slightly basally polarized (Figs 4A, S4, and S5).

Intestinal lipid absorption is a complex process that begins outside of the enterocytes with the breakdown of dietary triacylglycerols into free fatty acids. These are then imported into enterocytes, where they are re-assembled into triacylglycerols and incorporated into chylomicrons, which are secreted into the lymphatics. We found that the polarization of lipid absorption proteins followed this streamlined process (Figs 4A, S4, and S6). The pancreatic lipase (PNLIP), which initiates the extracellular breakdown of triglycerides, was strongly localized in the apical side. The fatty-acid transporters CD36 and FATP4 (*SLC27A4*), as well as the intestinal fatty acid-binding protein (I-FABP, *FABP2*) and the triacylglycerol assembly enzymes MGAT and DGAT were also apically biased. Chylomicron apolipoproteins exhibited diverse polarization patterns. APOA4 was apical, APOB-48 was balanced, and APOA1 was basal. These localization patterns could be explained by the role of APOA1 as both a late-incorporated chylomicron protein and a component of high density lipoprotein (HDL), which is exported basolaterally with ingested cholesterol [11,12]. Our polarization measurements therefore capture the streamlined process of carbohydrate and lipid absorption. Notably, these polarization trends remain unchanged after filtering out nuclear and matrix proteins, which are more abundant in the basal samples in our data (S7A Fig).

Our protein measurements revealed differences in the polarization patterns of ribosomal and mitochondrial proteins between humans and mice (Fig 4B). Whereas in mice, ribosomes were shown to localize apically and mitochondria basally [3], in humans, ribosomal proteins were slightly basal and mitochondria were slightly apical. We confirmed this finding by smFISH for *18S* rRNA and *MT-RNR1*/*mt-Nd6* mitochondrial RNA (Figs 4C, 4D, 4F, 4G, and S7B and S2 Table), and with RNA-seq (Fig 4E).

## Discussion

Intracellular mRNA localization has been shown to carry important roles in mammalian cells [13–20]. Localized translation of mRNA can confer optimality to cells by saving the energetic costs of protein transport and prevent toxicity that might be associated with ectopically translocated proteins. While prominent in elongated cells such as neurons, mRNA polarization has recently also been demonstrated in mammalian epithelial cells, specifically in a human intestinal cell line [21] and in the mouse small intestine [3]. To study the extent of mRNA and

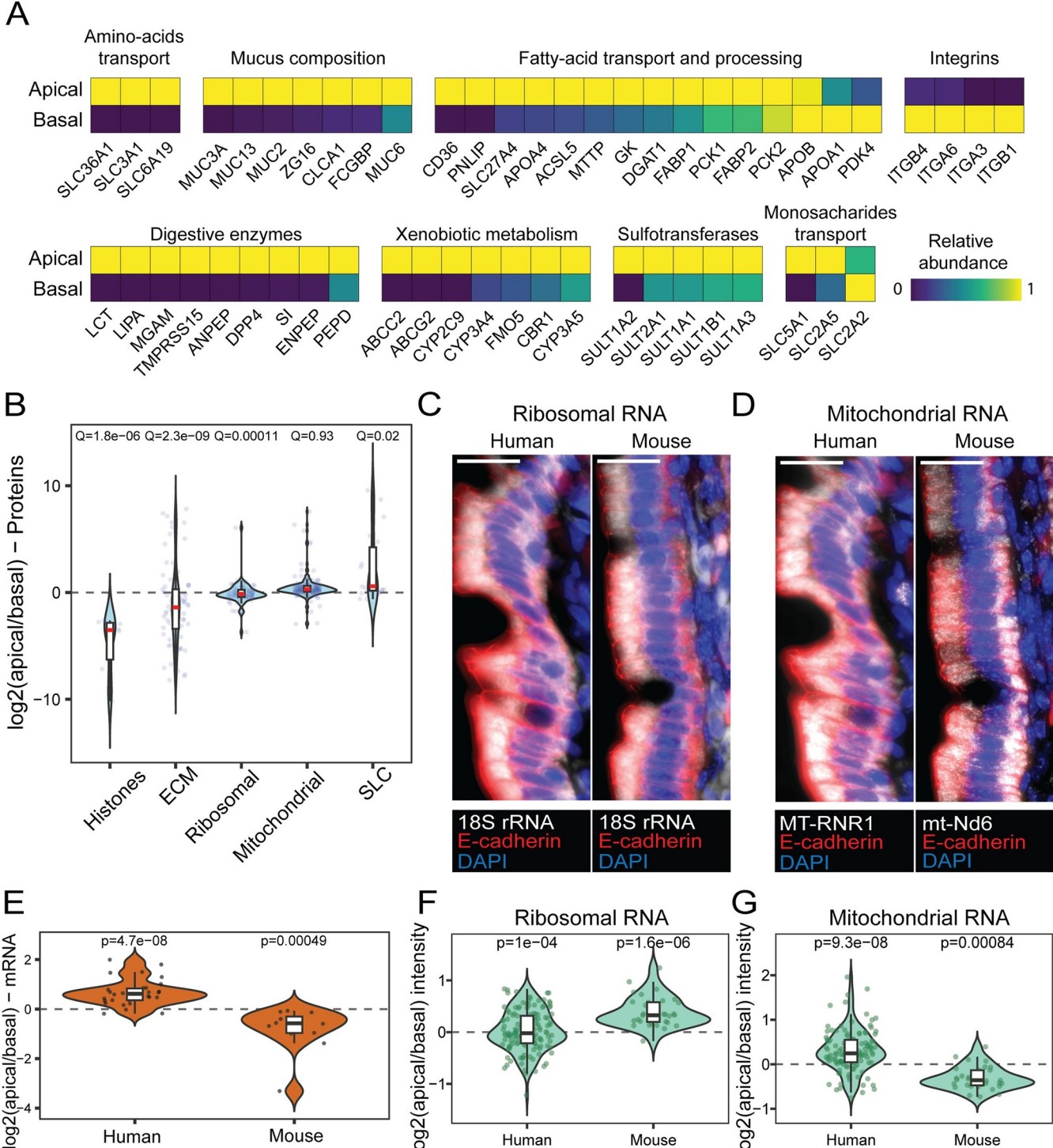

**Fig 4. Intracellular polarization of protein groups. (A)** Max-normalized abundance of proteins involved in nutrient processing and absorption on the apical and basal sides from LCM-proteomics data. **(B)** Log2(apical/basal) of protein groups based on LCM-proteomics. Q-values are based on Wilcoxon rank-sum test with Benjamini–Hochberg correction. ECM, extracellular matrix, SLC, solute carrier proteins. Horizontal bars are medians, boxes delineate the 25–75 percentiles. **(C, D)** smFISH of mitochondrial-RNA (C) and ribosomal-RNA (D). Red staining is anti E-cadherin antibody which stains basolateral epithelial membranes, blue staining is DAPI which stains DNA. Scale bar is 20 μm. Quantification shown in **(F and G)**. **(E)** The log2(apical/basal) of mitochondrially encoded genes with normalized expression above $10^{-4}$ in both human and mouse, based on RNA-seq. *P*-values are based on the paired Wilcoxon signed rank test. Horizontal bars are medians, boxes delineate the 25–75 percentiles. **(F, G)** Fluorescence intensity of smFISH stainings in the apical and basal sides of ribosomal (F) and mitochondrial (G) RNA. Each measurement consists of the median intensity across 3–5 epithelial cells from 8 patients and 3 mice. *P*-values

are based on the paired Wilcoxon signed rank test for each patient/mouse, combined by the Fisher method for multiple *p*-values (Methods). Horizontal bars are medians, boxes delineate the 25–75 percentiles. Data is available in S4 Table and S1 Data.

protein polarization in humans, we set out to characterize the human intestinal mRNA and protein apicome in vivo. We found that similar to mice, mRNA polarization is present in human intestinal epithelial cells and is highly correlated with the mRNA polarization patterns observed in mice. A notable example of a gene that showed different mRNA polarization between both species was *PIGR*, which showed basal polarization in mice and apical polarization in humans. The localization of Pigr mRNAs in mice remained constant in various conditions that modulate immune status, suggesting that the change is specie-dependent rather than state dependent. Future work could utilize the differentially polarized genes to explore potential mechanisms for specific mRNA trafficking. Notably, it has been observed that polarization of mRNAs exhibits dynamic changes following meal consumption in mice [3]. As human tissues procured during surgical interventions are generally obtained from individuals in a fasted state, it remains to be seen whether mRNA and protein polarization change with feeding.

Cells such as neurons, tend to localize mRNAs and corresponding proteins, with the aim of achieving fast response through localized translation [13,22–24]. We found that even in the much smaller enterocytes, there is a general correlation between the location of the mRNA and the corresponding protein. RNA polarization in human enterocytes might therefore serve to rapidly obtain proteins in their correct cellular compartment. Notably, the correlation between protein and mRNA polarization in humans contrasts the picture in mice, where mRNA and protein polarization do not generally correlate [3]. In mice, mRNA polarization has been shown to play a role in translational regulation. Ribosomes in the mouse intestinal epithelium are apically biased, resulting in higher translation rates of mRNAs that are polarized towards the apical cell side [3]. This suggests a potential mechanism to modulate translation rates at distinct villi zones via zonated switches in mRNA localization. For example, mRNAs that are biased towards the basal sides of cells at the villi bottoms, and towards the apical sides at the villi tips should yield higher translation rates per transcript at the tip. In our study, we did not identify zonal changes in polarization patterns in either humans or mice, suggesting that changes in polarization may not be a mechanism through which zonal modulation of translation rates is achieved. In line with this finding, a study that reconstructed villus zonation profiles of enterocyte mRNAs and proteins in mouse suggested that translation rates are largely constant along the villus axis [25].

We found that the distribution of the ribosomes across the apical/basal compartments in humans is uniform (Fig 4B, 4C, and 4F), in contrast to mice. Despite the uniform distribution, we found a correlation between the polarization of mRNAs and their translation rates (S2A Fig). This discrepancy could be explained by the presence of shorter, more abundant mRNAs, in the apical side—both of which are closely linked to translation rates [26,27]. Additionally, intracellular localization of mRNAs and proteins, including ribosome components, might be influenced by feeding status, a phenomenon that would be interesting to explore in future studies.

This study mapped the apical and basal polarization of mRNAs and proteins in the healthy human small intestinal epithelium. The basal compartment in the epithelial cells was at the resolution limit of the LCM and included residual nuclear fragments and consequently, basal polarization of nuclear proteins (Fig 4B). Despite this, the main conclusions of the study, including the streamlined polarization of enterocyte-specific proteins remained invariable upon removal of nuclear and stromal proteins (S7A Fig). This suggests that our approach for combined LCM-based transcriptomics and proteomics could be applied to characterize the

apicome of other human epithelial tissues, such as various glands, reproductive organs, kidney, colon, gallbladder, and stomach. Additionally, cell polarity is often perturbed in cancer and other gut diseases [28]. Future apicome studies could assess changes in cell polarity resulting from gut diseases, such as Crohn's and Celiac disease, as well as age and diet.

In summary, our map of mRNA and protein polarization provides an important resource for understanding human intestinal epithelial cell biology in vivo.

## Materials and methods

### Experimental methods

**Human tissues preparations.** Eight proximal jejunums were collected from patients undergoing pancreaticoduodenectomy surgery for the indications of pancreatic ductal adeno-carcinoma (PDAC) or intraductal papillary mucinous neoplasm (IPMN) (S1 Table). All operations were performed in Sheba Medical Center, Israel and approved by the Sheba Medical Center Helsinki committee (Helsinki approval number 866521SMC). Samples were taken only in cases where no pathology was observed in the small intestine. Human small intestinal tubes from the duodenum–jejunum junction (ligament of Treitz) were resected and longitudinally opened to expose the mucosal surface. Tissues were gently washed in phosphate-buffered saline (PBS) and mucosal sheets were sectioned by performing mucosectomy. Sheets were rolled to a "Swiss roll" and embedded in OCT (Tissue-Tek) and kept at −80˚C. Additionally, sheets were fixed in cold 4% formaldehyde (FA) in PBS for 3 h at 4˚C followed by overnight fixation in 4% FA + 30% sucrose at 4˚C while revolving prior to OCT embedding.

### Animal experiments and tissue preparations

Mouse experiments were approved by the Institutional Animal Care and Use Committee of the Weizmann Institute of Science (approval number 13000419–2) and performed in accordance with institutional guidelines. C57BL/6 Male mice aged 2 to 5 months, housed under regular 12 h light-dark cycle and fed ad libitum were used in the experiments.

For the antibiotic treatment, mice were given a combination of the following antibiotics for 4 weeks, vancomycin (1 g/l), ampicillin (1 g/l), kanamycin (1 g/l), and metronidazole (1 g/l) in their drinking water [29–31], all antibiotics were obtained from Sigma Aldrich. For LPS treatment—LPS was dissolved in sterile PBS. Mice were injected 20 μg/kg body weight LPS intraperitoneally, mice were sacrificed 12 h after the injection. Overall, 2 mice from each condition and 5 controls were used in the study.

Mice sacrificed by cervical dislocation and the jejunums were harvested and washed with cold PBS. The proximal half of each jejunum was lateralized and spread onto a Whatman paper, embedded in OCT, and kept at −80˚C. The other half of each jejunum was fixed in cold 4% FA in PBS for 3 h at 4˚C followed by overnight fixation in 4% FA + 30% sucrose at 4˚C while revolving. The jejunums were rolled to a "Swiss-roll," embedded in OCT, and kept at −80˚C.

### Single molecule fluorescence in situ hybridization

Probe libraries were designed using the Stellaris FISH Probe Designer Software (Biosearch Technologies, Inc., Petaluma, California, United States of America). Probe libraries (S5 Table) were coupled to Cy5, TMR, or Alexa594. Cryosections (6 μm) of fixed mouse or human intestines were mounted on poly-L-lysine pre-coated coverslips and used for probe hybridization, as previously described [32]. Briefly, sections were fixed for 15 min in 4% FA in PBS, washed with PBS, and incubated for 2 h in 70% ethanol in 4˚C. Sections were washed with 2× SSC

(Ambion AM9765) for 5 min, then permeabilized for 10 min with proteinase K (10 mg/ml, Ambion AM2546) followed by 2 washes with 2× SSC (Ambion AM9765) for 5 min. Tissues were incubated in wash buffer (20% Formamide Ambion AM9342, in 2× SSC) for 10 min in a 30°C incubator. Tissues were incubated with hybridization mix (10% Dextran sulfate Sigma D8906, 20% formamide, 1 mg/ml *E. coli* tRNA Sigma R1753, 10% 20× SSC, 0.02% BSA Ambion AM2616, 2 mM Vanadylribonucleoside complex NEB S1402S, in Rnase-free water) mixed with probes and FITC-conjugated anti-CDH1 antibody (1:100, BD Biosciences cat. 612131) overnight in 30°C. For negative control staining—hybridization mix was used without addition of probes. After the hybridization, tissues were washed with wash buffer for 30 min in 30°C, and with wash buffer with DAPI (50 ng/ml, Sigma-Aldrich, D9542) for 5 min. Additional wash with 2× SSC was performed, and coverslips were mounted with ProLong Gold antifade reagent (Invitrogen P36934). All images were performed on a Nikon-Ti-E inverted fluorescence microscope using the NIS element software AR 5.02.03. Images were acquired using 100× magnification. All smFISH experiment were performed on tissues from at least 3 patients from multiple fields of view.

### Laser capture microdissection

Cryosections (10 μm) of fresh-frozen mouse or human intestines were mounted on polyethylene-naphthalate membrane-coated glass slides (Zeiss, A4151909081000), air-dried for 20 s at room temperature, washed in 70% ethanol for 30 s, incubated in Rnase-free water for 30 s, followed by staining with HistoGene Staining Solution for 20 s (Thermo Fisher Scientific, KIT0401). The stained sections were dehydrated with subsequent 30 s incubations in water, and then in 70%, 95%, and 100% ethanol and air dried for 4 min before microdissection. Tissue sections were microdissected on a UV laser-based PALM Microbeam (Zeiss). The system makes use of a pulsed UV laser that cuts the tissue at indicated marks with minimal damage to surrounding cells; the cutting was performed with the following parameters: PALM 40X lens, cut energy 43 (1–100), cut focus 52 (1–100). Tissue fragments were catapulted and collected in 0.2 ml adhesive cap tubes (Zeiss, A4151909191000) with these settings: LPC energy 68 (1–100), LPC focus 50 (1–100). For RNA-seq, the cutting time of each slide was limited up to 80 min, and buffer RLT from QIAGEN with 0.25 M Dithiothreitol was added and immediately frozen on dry ice. For MS, 5% SDS, 50 mM Tris (pH 8) was added before freezing.

### RNA-seq

RNA-seq Library preparation was performed following mcSCRBseq protocol ([33]). The cDNA was pre-amplified with 17 cycles. Up to 0.6 ng of the amplified cDNA was converted into sequencing library with the Nextera XT DNA Library kit (Illumina, FC131-1024), according to supplier protocol. Quality control of the resulting libraries was performed with an Agilent High Sensitivity D1000 ScreenTape System (Agilent, 5067–5584). Libraries were loaded with a concentration of 1.8 nM and sequenced on a Novaseq 6000 (Illumina) with the following cycle distribution: 8 bp index1, 26 bp read1, 8 bp index2, and 66 bp read2. Illumina output files were demultiplexed and aligned to the genome with UTAP [34], with CUTADAPT with default parameters. A total of 32 human samples from 8 patients were aligned to the HG38 genome (Gencode_V34_24.3.2020), and 18 mouse samples from 5 mice were aligned to the mm10 genome (Gencode_vM25_24.3.2020).

### Proteomics of LCM samples

LCM samples from 4 patients were lysed and digested with trypsin using the S-trap method [35]. Lysates in 5% SDS in 50 mM Tris-HCl were incubated at 96°C for 5 min, followed by 6

cycles of 30 s of sonication (Bioruptor Pico, Diagenode, USA). Proteins were reduced with 5 mM dithiothreitol and alkylated with 10 mM iodoacetamide in the dark. Phosphoric acid was added to the lysates to a final concentration of 1.2%, and 90:10% methanol/50 mM ammonium bicarbonate and then added to the samples. Each sample was then loaded onto S-Trap micro-columns (Protifi, USA). Samples were then digested with 250 ng trypsin for 4 h at 37˚C. The digested peptides were eluted using 50 mM ammonium bicarbonate; trypsin was added to this fraction and incubated overnight at 37˚C. Two more elutions were made using 0.2% formic acid and 0.2% formic acid in 50% acetonitrile. The 3 elutions were pooled together and vacuum-centrifuged to dry. Samples were kept at −80˚C until analysis.

## Liquid chromatography

ULC/MS grade solvents were used for all chromatographic steps. Each sample was loaded using split-less nano-Ultra Performance Liquid Chromatography (10 kpsi nanoAcquity; Waters, Milford, Massachusetts, USA). The mobile phase was: (A) H2O + 0.1% formic acid; and (B) acetonitrile + 0.1% formic acid. Desalting of the samples was performed online using a reversed-phase Symmetry C18 trapping column (180 μm internal diameter, 20 mm length, 5 μm particle size; Waters). The peptides were then separated using a T3 HSS nano-column (75 μm internal diameter, 250 mm length, 1.8 μm particle size; Waters) at 0.35 μl/min. Peptides were eluted from the column into the mass spectrometer using the following gradient: 4% to 30%B in 105 min, 30% to 90%B in 10 min, maintained at 90% for 7 min and then back to initial conditions.

## Mass spectrometry

The nanoUPLC was coupled online through a nanoESI emitter (10 μm tip; Fossil, Spain) to a quadrupole orbitrap mass spectrometer (Exploris 480, Thermo Scientific) using a FlexIon nanospray apparatus (Thermo Scientific). Data was acquired in data-dependent acquisition (DDA) mode, with a cycle time limit of 2 s. MS1 resolution was set to 120,000 (at 200 m/z), mass range of 380–1,500 m/z, AGC of 200% and maximum injection time was set to 50 msec. MS2 resolution was set to 15,000, quadrupole isolation 1.4 m/z, AGC of 150%, dynamic exclusion of 40 sec and maximum injection time of 150 msec. Raw data was processed with Max-Quant (v2.0.1) [36]. The data was searched with the Andromeda search engine against the human proteome database appended with common lab protein contaminants and the default modifications. Quantification was based on the intensity-based absolute quantification (iBAQ) method, based on unique peptides, missing values were imputed from a low, random distribution.

## Computational methods

**Compartment size measurement.**   Segments from the apical and basal sides of several epithelial cells from spatially distinct villi zones from and spatially distant villi were manually measured using imageJ [37] for each of the 8 patients. The annotations were performed with nuclear and membrane staining. Log2(apical/basal) ratios were calculated for each cell and visualized with violin plot in ggplot 2 3.5.0 [38], and Wilcoxon rank-sum was performed using the ggpubr 0.6.0 [39] (S1A Fig).

## RNA-seq analysis pipeline

RNA-seq results yielded alignment of 17,947 and 17,019 genes for human and mouse samples, respectively. In mouse data, one low-quality sample was removed due to excessively low

unique molecular identifier (UMI) counts (below 10,000). Non-protein-coding genes were removed, based on BioMart [40] gene_biotype values.

Data were normalized by dividing each value in a sample by the sum of UMIs in the sample. For principal component analysis (PCA), samples with more than 20,000 UMIs were used, and genes with mean expression $>10^{-4}$ were scaled, and PCA was performed.

Pooled apical bias for each gene was calculated by geometric mean of the apical/basal ratios of all patients/mice, as follows:

$$Apicome\ score = \frac{1}{n} * \sum_{i=1}^{n} \log_2 \frac{A_i + pn}{B_i + pn}$$

Where is number of patients/mice ($n = 8$ for human and $n = 5$ for mouse), $A$ and $B$ are the normalized apical and basal expression, respectively, and $pn$ is pseudo-number which is the minimal normalized expression across all samples and all genes, which is not 0. This value is added to avoid division by zero and consequently infinite or non-defined values when expression levels are low, without skewing the ratios. $P$-values for apical/basal bias and for changes between top/bottom of villi were calculated using exact paired sign-rank test with two-sided alternative, on highly expressed genes (mean normalized expression $>10^{-5}$). Q-values were calculated using the Benjamini–Hochberg method.

Gene set enrichment analysis (GSEA) was performed on the log2(apical/basal) values of all genes, using Fgsea 1.28.0 [41] package, fgseaSimple function, with the "hallmark" database.

## Correlation of RNA-seq with previously published data

Spearman correlation was performed to assess the relationship between log2(apical/basal) and various biological parameters. The parameters included data from Schwanhäusser and colleagues [26] and Harnik and colleagues [25]. mRNA expression and protein abundance are from the current study, calculated as the mean expression of the normalized LCM RNA-seq and proteomics.

Spearman's correlation was also calculated on the log2(apical/basal) from Moor and colleagues [3] in comparison to the log2(apical/basal) from the current study, on significant and highly expressed genes (q-value in both data sets $<0.1$, max_expression $>10^{-5}$) using the ggpubr 0.6.0 [39] and ggplot 2 3.5.0 [38] libraries (S1B Fig).

## Comparison of mouse and human RNA-seq

RNA-seq analysis of both species was merged. Genes were renamed based on orthology table from Ensembl (V.109), obtained with BioMart [40]. Spearman correlation analysis was performed based on log2(apical/basal) as described in the RNA-seq analysis section. For each gene, Wilcoxon rank-sum test was performed for the human samples and the mouse samples. $P$-values were corrected for multiple hypotheses using the Benjamini–Hochberg method. Genes were determined as significant based on the q-value of the inter-species q-value $<0.2$ and also the minimal species-specific polarization q-value $<0.1$. For mitochondrial apicome bias—one-sample, two-sided signed rank test was performed on mitochondrial genes with normalized expression $>10^{-4}$.

## smFISH quantification of gene expression

smFISH images were analyzed by manual segmentation of apical and basal compartments of few cells and identification of dots using imageM [32]. For each gene, multiple FOVs from distinct 3 villi per patient, across 2 to 3 patients were quantified. Spot intensity was normalized by

the segmented area and the log2(apical/basal) ratio was calculated for each paired segmentation. The mean and standard error of the mean were calculated for the log2(apical/basal) ratios.

## Proteomics data analysis

Suspected contaminants, as well as proteins with number of unique peptides lower than 2 were removed. The minimal iBAQ value was subtracted from all values, and each iBAQ value was divided by the sum of iBAQ of its sample. Low-quality samples, with number of identified proteins <1,000 were filtered out. Differential gene expression analysis (DGE) was performed using Seurat 5.0.1 [42] FindMarkers function with DESeq2 method [43]. Log2(apical/basal) was defined as $log_2\left(\frac{apical\ expression\ in\ tip+pn}{basal\ expression\ in\ tip+pn}\right)$, where $pn = 10^{-6}$. ECM proteins were assigned based on Matrisome database [8], mitochondrial proteins were assigned by gene description from Ensembl (V.109), obtained with BioMart [40] and metabolic proteins were obtained from Burclaff and colleagues [44]. For each group, $p$-value was calculated with two-sided Wilcoxon rank sum test (proteins in group versus all the rest of the proteins), and Benjamini–Hochberg correction was performed.

For heatmaps visualization—normalized abundance of each protein was defined as follows:

$$Apical = \frac{Apical}{MAX(Apical, Basal)}\ Basal = \frac{Basal}{MAX(Apical, Basal)}.$$

And heatmaps were visualized using ggplot2 3.5.0 [38]. KEGG pathways were visualized with KEGGREST 1.42.0 [45] and pathview libraries [46].

The analysis was repeated, focusing exclusively on cytoplasmic proteins. Subcellular localization data was obtained from the Human Protein Atlas [47] and alternative protein names were retrieved from UniProt [48]. Proteins identified solely as nuclear based on subcellular location gene annotation, as well as ECM proteins according to the Matrisome [8], were excluded. Following this filtration, data was renormalized by dividing each value by the sum of values in the sample.

## mRNA and protein correlation in cellular polarization

The log2(tip/base) ratios from the average of RNA-seq apical and basal samples were merged with the corresponding proteomics apical samples. Spearman's correlation was then calculated and visualized for relatively highly expressed genes (normalized expression $>10^{-4}$) using the ggpubr 0.6.0 [39] and ggplot2 3.5.0 [38] libraries. Additionally, Spearman's correlation was calculated on the log2(apical/basal) ratios of RNA-seq and proteomics on relatively highly expressed genes (normalized expression $>10^{-5}$).

Epithelial-specific genes were classified based on integration of Elmentaite and colleagues [49] and Burclaff and colleagues [44] single-cell RNA-seq data and cell type classifications. The mean of cells from each cell type was calculated and then the signature-matrix was normalized where each value was divided by the sum of values in the cell type. Enterocyte-specific genes were classified based on the following:

$$Epithelial\ specific = \begin{cases} True\ if\ \frac{E+pn}{N+pn} > 2, E > 1*10^{-5} \\ False\ Otherwise \end{cases}.$$

Where $E$ is the expression in mature enterocytes, $N$ is the maximal expression across all non-enterocyte cell types, and $pn$ is the minimal non-zero value present in the signature-matrix.

This value is added to avoid division by zero and consequently infinite or non-defined values when expression levels are low, without skewing the ratios.

### Mitochondrial/ribosomal intensity quantification of smFISH images

Segments from the apical and basal sides of 3 to 5 epithelial cells from spatially distant villi were manually segmented on nuclear and membrane staining using ImageJ [37] for each of the 8 patients and of 3 mice (S2C Fig). The median intensity for mitochondrial and ribosomal stainings were quantified in both sides, and the corresponding median intensity of the background was subtracted. Ratio of apical/basal background-subtracted intensities was calculated, and 2 signed-rank tests were performed on each patient, one test with left-tail and one with right-tail hypothesis. The minimal *p*-value of both tests was combined across the 8 patients and across the 3 mice using Fisher's method.

## Supporting information

**S1 Fig.** **(A)** Top–Immuno-fluorescence staining of E-cadherin, used for measurement of apical (yellow) and basal (red) sides of the epithelial cells. Scale bar 20 μm. Bottom–quantification of measurements from 8 patients, 3 FOV per patient, 3 measurements per FOV. Boxplots show the medians and 25–75 percentiles, *p*-value is Wilcoxon rank-sum test. Data is available in S2 Table and S1 Data. **(B)** Number of unique molecules identifiers (UMIs) across apical and basal samples of LCM RNA-seq of human proximal jejunums. Boxplots show the medians and 25–75 percentiles, *p*-value is Wilcoxon rank-sum test. Data is available in S3 Table and S1 Data. **(C)** Principal component analysis (PCA) of human LCM RNA-seq, samples colored by cellular compartment (left), by villus zone (mid) or by patient (right). Only samples with more than 20,000 UMIs and genes with mean normalized expressions of more than $10^{-4}$ are included. Data is available in S3 Table and S1 Data. **(D)** log2(apical/basal) from LCM RNA-seq of 5 mice. Apical/basal classification based on previously published data of mouse apicome [3]. Shown are genes that are significantly polarized in Moor and colleagues data set (q-value < 0.1). Horizontal bars are medians, boxes delineate the 25–75 percentiles. Data is available in S3 Table and S1 Data. **(E)** smFISH staining of SLC5A1 and a staining without a probe. Scale bar is 20 μm. **(F)** Spearman correlation between the log2(apical/basal) in LCM RNA-seq and quantification of dot intensity of smFISH. In smFISH, intensity of dots (normalized to area), from multiple fields of view, from 2 to 3 patients were quantified. Error bars are standard errors of the means of the log2(apical/basal) from all samples or measurements. SmFISH quantification and RNA-seq data are available in S2 and S3 Tables, respectively, and S1 Data. (TIF)

**S2 Fig.** **(A)** Spearman correlation of log2(apical/basal) ratios from LCM RNA-seq and different gene parameters, taken from Schwanhäusser and colleagues [26] and Harnik and colleagues [25]. mRNA expression and protein abundance values are the averages of normalized expression data from the current study. **(B)** Gene set enrichment analysis (GSEA) results based on log2(apical/basal) of LCM RNA-seq. Only pathways with q-value < 0.1 are shown. Data is available in S3 Table and S1 Data. **(C)** Spearman correlation between the log2(apical/basal) in human data, in bottom and tip of villi. Only genes with normalized expression $>10^{-4}$ are included. Data is available in S3 Table and S1 Data. **(D)** smFISH staining of Pigr in the jejunums of mice in different conditions. In red–E-cadherin immunofluorescence, in blue–DAPI. Scale bar is 20 μm. (TIF)

**S3 Fig. (A)** Spearman correlation of the log2(tip of villi/base of villi), of LCM RNA-seq and proteomics. Only genes with normalized expression in both data sets $>10^{-4}$ are included. RNA-seq and proteomics data are available in S3 and S4 Table, respectively. **(B)** Protein log2 (apical/basal), where proteins are classified based on the coverage in mouse brush border isolation from McConnell and colleagues [6]. Horizontal bars are medians, boxes delineate the 25–75 percentiles. **(C)** Bottom–number of genes in each state, depending on the polarization of the mRNA and the corresponding protein. *P*-values are based on hypergeometric tests. Top–distribution of concordant and discordant genes, colors correspond to the colors in the bottom panel. Data is available in S4 Table and S1 Data. **(D)** Spearman correlation between the log2 (apical/basal) of mRNAs and the corresponding log2(apical/basal) of proteins, Only epithelial specific genes are included (Methods). RNA-seq and proteomics data are available in S3 and S4 Tables, respectively, and in S1 Data.
(TIF)

**S4 Fig. Max-normalized abundance of proteins involved in nutrient processing and absorption on the apical and basal sides from LCM-proteomics data.** Horizontal bars are medians, boxes delineate the 25–75 percentiles. Data is available in S4 Table and S1 Data.
(TIF)

**S5 Fig. KEGG carbohydrate digestion and absorption pathway.** Color is determined by the log2(apical/basal) of LCM-proteomics, where all values above 1 are rounded to 1, and all values below −1 are rounded to −1. Visualized with Pathview [46].
(TIF)

**S6 Fig. KEGG fat digestion and absorption pathway.** Color is determined by the log2(apical/basal) of LCM-proteomics, where all values above 1 are rounded to 1, and all values below −1 are rounded to −1. Visualized with Pathview [46].
(TIF)

**S7 Fig. (A)** Max-normalized abundance of cytoplasmic proteins involved in nutrient processing and absorption on the apical and basal sides of cytoplasmic proteins from LCM-proteomics data. The data is internally normalized after filtering out nuclear and ECM proteins (Methods). **(B)** smFISH image of mt-Nd6 in mouse jejunum, as example of segmentation and quantification of the intensity. Scale bar 20 μm.
(TIF)

**S1 Table. Patients' information used in the study.**
(XLSX)

**S2 Table. Image analysis data.** Compartment size measurements, quantification of smFISH dots intensity, and quantification of mtRNA and rRNA intensity in humans and mice.
(XLSX)

**S3 Table. Raw and processed RNA-seq data of epithelial apical and basal segments isolated from the top and bottom of human and mouse villi from 8 patients and 5 mice.** Table also contain comparison between human and mouse, and enterocyte specific genes. The table also contains GSEA results of human RNA-seq.
(XLSX)

**S4 Table. Raw and processed mass spectrometry proteomics data of epithelial apical and basal segments isolated from the top and bottom of human villi from 4 patients.**
(XLSX)

**S5 Table. smFISH probes used in the study.**
(XLSX)

**S1 Data. Source data for all figures presenting numerical data.**
(XLSX)

## Author Contributions

**Conceptualization:** Roy Novoselsky, Yotam Harnik, Shalev Itzkovitz.

**Data curation:** Roy Novoselsky, Yotam Harnik, Oran Yakubovsky, Keren Bahar Halpern, Niv Pencovich, Ido Nachmany.

**Formal analysis:** Roy Novoselsky.

**Funding acquisition:** Shalev Itzkovitz.

**Investigation:** Roy Novoselsky, Yotam Harnik, Oran Yakubovsky, Corine Katina, Yishai Levin, Keren Bahar Halpern, Niv Pencovich, Ido Nachmany, Shalev Itzkovitz.

**Project administration:** Shalev Itzkovitz.

**Software:** Roy Novoselsky.

**Supervision:** Shalev Itzkovitz.

**Validation:** Roy Novoselsky.

**Visualization:** Roy Novoselsky.

**Writing – original draft:** Roy Novoselsky, Shalev Itzkovitz.

**Writing – review & editing:** Roy Novoselsky, Yotam Harnik, Oran Yakubovsky, Shalev Itzkovitz.

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
