## [Editor Report · Decision Letter 0]

2 May 2024

Dear Dr Itzkovitz, 

Thank you for submitting your manuscript entitled "Intra-cellular polarization of RNAs and proteins in the human small intestinal epithelium" for consideration as a Short Report by PLOS Biology.

Your manuscript has now been evaluated by the PLOS Biology editorial staff as well as by an academic editor with relevant expertise and I am writing to let you know that we would like to send your submission out for external peer review.

Once your full submission is complete, your paper will undergo a series of checks in preparation for peer review. After your manuscript has passed the checks it will be sent out for review. To provide the metadata for your submission, please Login to Editorial Manager (https://www.editorialmanager.com/pbiology) within two working days, i.e. by May 06 2024 11:59PM.

Kind regards,

Ines

--

Ines Alvarez-Garcia, PhD

Senior Editor

PLOS Biology

---

## [Decision Letter · Decision Letter 1]

22 Jul 2024

Dear Dr Itzkovitz,

Thank you for your patience while your manuscript entitled "Intra-cellular polarization of RNAs and proteins in the human small intestinal epithelium" was peer-reviewed at PLOS Biology as a Short Report. Please also accept my apologies again for the delay in sending you our decision. Your manuscript has been evaluated by the PLOS Biology editors, an Academic Editor with relevant expertise, and by three independent reviewers.

The reviews are attached below. As you will see, although the reviewers find the work potentially interesting, they have also raised several important concerns. Reviewer 1 misses functional validation of the physiological relevance of the findings and notes that there are significant differences in the polarisation patterns between the two species. In addition, this reviewer asks whether there is a correlation between protein turnover rates and RNA polarisation, as well as between mRNA stabilisation and localisation. Reviewer 2 requests an analysis of the villus tip and villus base domain separation in humans, which is missing, and also adding the polarity of SLCSA1 performing IF in mouse intestine along with other clarifications. Reviewer 3 raises a fundamental issue regarding the LCM proteomic results, arguing that the approach mainly compares a fraction highly enriched in cytoplasm, or apical part, with a fraction highly enriched in nuclear content, or basal side, and that this level of resolution is very limiting. We don’t learn if there are any cytosolic/membrane proteins enriched in the basal fraction, beyond ECM, or if most of these correspond to nuclear proteins.

Based on their specific comments and following discussion with the Academic Editor, it is clear that a substantial amount of work would be required to meet the criteria for publication in PLOS Biology. However, given the interest of your study, we would be open to inviting a comprehensive revision of the study that thoroughly addresses all the reviewers' comments. In addition, the Academic Editor would like you to comment on the difference in the subcellular distribution of PIGR, which is apical in humans, but not in mice. Since the expression is regulated by cytokines in response to pathogenic stimuli, one may wonder whether this change in protein localisation is state-dependent, rather than species-specific.

Given the extent of revision that would be needed, we cannot make a decision about publication until we have seen the revised manuscript and your response to the reviewers' comments. Your revised manuscript would need to be seen by the reviewers again, but please note that we would not engage them unless their main concerns have been addressed.

We appreciate that these requests represent a great deal of extra work, and we are willing to relax our standard revision time to allow you 6 months to revise your study. Please email us (plosbiology@plos.org) if you have any questions or concerns, or envision needing a (short) extension.

**IMPORTANT - SUBMITTING YOUR REVISION**

3. Resubmission Checklist

a) *PLOS Data Policy*

b) *Published Peer Review*

d) *Blurb*

Please also provide a blurb which (if accepted) will be included in our weekly and monthly Electronic Table of Contents, sent out to readers of PLOS Biology, and may be used to promote your article in social media. The blurb should be about 30-40 words long and is subject to editorial changes. It should, without exaggeration, entice people to read your manuscript. It should not be redundant with the title and should not contain acronyms or abbreviations. For examples, view our author guidelines: https://journals.plos.org/plosbiology/s/revising-your-manuscript#loc-blurb

Sincerely,

Ines

--

Ines Alvarez-Garcia, PhD

Senior Editor

PLOS Biology

Reviewers' comments

Rev. 1:

The manuscript "Intra-cellular polarization of RNAs and proteins in the human small intestinal epithelium" deciphers and compares the spatial distribution of mRNA molecules and proteins in mouse and human intestinal epithelial cells along the villus axis. By utilizing laser-capture microdissection, the study generates high-throughput transcriptome and proteome data from intracellular apical and basal compartments, creating a novel apical-basal polarization profile (apicome) for the human intestinal epithelium. The authors demonstrate clear polarization of mRNA molecules and proteins in the human intestinal epithelial cells, similar to patterns observed in mice. This study also highlights significant differences in polarization patterns between the two species, including a lack of ribosomal protein polarization in humans, suggesting that the previously identified RNA polarization-dependent machinery regulating translation rate in mice does not exist in humans. The characterized apicome of proteins involved in nutrient transport and metabolic processes aligns with their known in vivo functions, underscoring the significance of these findings for understanding human epithelial cell functions. Additionally, the comparative study is valuable for further investigation on species-specific cellular mechanisms and their evolutionary implications. While this manuscript is well written, it is an incremental advance in our understanding of intestinal biology. Additional analyses would help strengthen the significance.

1. The manuscript demonstrates RNA and protein polarization within mouse and human intestinal epithelial cells but did not perform any functional validation of physiological relevance. For example, it would be interesting to investigate the correlation between protein turnover rates and RNA polarization, as well as between mRNA stabilization and localization. (Why is RNA polarization required when ribosomes are evenly distributed in human epithelial cells?) Are there certain classes of proteins/genes that have localized mRNA/protein?

2. The authors conclude that "proteins and mRNAs in human enterocytes are generally localized in their apical/basal cellular compartments". However, the Spearman correlations in fig.3C and fig.S2B (R=0.13 and R=0.32) are not strong enough to support this conclusion. It would also be helpful to show the percentage of polarized RNAs that share the same localization as their encoded proteins.

3. The expression levels of certain proteins are different between the villus tip and bottom. It would be interesting to examine the differences in RNA polarization at various villus regions for those proteins to find possible relationship between translation efficiency and RNA localization.

4. The statistical analysis in fig.4E-G requires more detailed explanation.

5. In fig.S1A, the clustering in the principal component analysis of human LCM-RNAseq data is not clear and distinct. Only apical compartment samples show robust clustering.

Rev. 2:

This is one of a number of high impact studies from the Itzkovitz laboratory and studies apical and basolateral polarity of mRNA, mitochondria and proteins comparing human and mouse. There are several concerns with the study although except for the first point they are minor:

1) The methodology allows both apical and basolateral separation as well as villus tip and villus base separation. However, striking is lack of analysis of the villus tip and villus base domain separation in human that was so well done in the mouse; with the hugely important difference between mRNA and protein zonation. It is reasonable to ask that this analysis be added to the manuscript, although this would be far less in detail than was provided for the mouse, it would still be impactful.

2) Add some comment why think chemotherapy in human patients not affecting result.

3) The location of SLC5A1 mRNA in mouse being partially basal is on of the surprising observations; please provide the polarity of the protein by IF in mouse intestine. Also calling SLC5A1 discordant for human as in Fig 4D is not correct.

4) Do not see PNLIP in Fig 4A which would be interesting given that it is thought to sticks apically.

5) In Methods for RNA-seq formula; explain why pn added (to both numerator and denominator).

Rev. 3:

SUMMARY AND STRENGHTS:

In this manuscript, Novoselsky and colleagues perform transcriptomics and proteomics analysis of basal and apical portion of small intestinal epithelium from human and mice and validated the findings using single-molecule FISH and immunohistochemistry (obtained from human protein atlas website). They find transcript and proteins with apical or basal localization polarization, which sometimes correlates for the same gene. They suggest presence of a streamlined " nutrient and transport and processing" in intestinal cells. The strength of the paper is the use of human tissues and the comparison with mouse and the systematic approach allowed by using RNA-Seq and Mass Spectrometry.

ISSUES:

However, there are several issues with the conceptual design of the paper and the methods employed.

There is limited usefulness in the assessment of apical/basal localization of mRNA and proteins, especially due to the use of laser-capture microdissections method. Due to the basal localization of the nuclei (also noted by the authors in the results section) and the limited abundance of cytoplasm in the basal side, the use of LCM will results essentially in the comparison of a fraction highly enriched in cytoplasm (the apical part) with a fraction highly enriched in nuclear content (basal side). This is reflected in their data as most of the proteins show apical bias except histones and ribosomes and mitochondria which are known to reside in proximity to the nucleus. Therefore, this analysis is of limited usefulness to really assess cellular localization of proteins. This is also reflected in the limited novelty provided by the paper. This is a crucial point that should be clearly mentioned in the discussion.

Some of the methodologies/data representation need to be improved:

In smFISH experiments: there should be noted how many patients were analysed and how many FOVs were analysed for each patients. Quantification of the signal in apical VS basal localization should also be carried out. Negative controls should also be shown.

It seems that there are multiple repeats of the same image: -CDH1 staining from Fig 3D is a repeat of 2B (image rotated), SLC5A 2A is a repeat of 2D and 3D, this should be replaced with different representative images.

In figure 2D: ApoB FISH is also shown but not mentioned in the text, therefore the logic of showing this is unclear.

Data in figure 4a should also been show as box plots, histograms or violin plot showing single-point data and error bars.

Number of patients analysed for each figure/panel should also be reported in the legends for ease.

When handful of genes are highlighted (ie Fig2C, 3A and C) it should be specified that these are selected by authors not statistically.

The concept of apicome is not explained clearly and generally confusing. In line 72 for instance it would read more clearly: "the samples clustered by their apical or basal origin and villi zone rather than..." is the apicome only apical protein? Is the comparison of apical and basal? The term should be changed or use it differently in the text.

It would also be useful to expand on the "apicome score" for instance by indicate that positive numbers imply apical polarization and negative number a basal polarization. Also, in figure this is expressed as log2(apical/basal) rather than with the term "apicome score", perhaps unifying the terminology would make manuscript clearer.

---

## [Decision Letter · Decision Letter 2]

9 Oct 2024

Dear Dr Itzkovitz,

Thank you for your patience while we considered your revised manuscript entitled "Intra-cellular polarization of RNAs and proteins in the human small intestinal epithelium" for publication as a Short Report at PLOS Biology. This revised version of your manuscript has been evaluated by the PLOS Biology editors, the Academic Editor and by the three original reviewers.

Based on the reviews (attached below), we are likely to accept this manuscript for publication, provided you satisfactorily address the data and other policy-related requests stated below.

*We routinely suggest changes to titles to ensure maximum accessibility for a broad, non-specialist readership, and to ensure they reflect the contents of the paper. Please consider this suggestion (or similar) to improve the title:

"Intra-cellular polarization of RNAs and proteins in the human small intestinal epithelium can differ from patterns observed in mice"

**In addition, we think that using the word 'patients' in the Abstract without specifying the context is a bit confusing, as their intestines are normal. Please either use 'individuals' instead, or explain why they are referred as patients. 

We expect to receive your revised manuscript within two weeks. 

*Published Peer Review History*

*Press*

Sincerely,

Ines

--

Ines Alvarez-Garcia, PhD

Senior Editor

PLOS Biology

ETHICS STATEMENT:

Thank you for including the ethics statement. Please also provide the approval number.

Fig. 2A, C; Fig. 3A, C; Fig. 4B, E-G; Fig. S1A-D, F; Fig. S2B, C; Fig. S3A, B, D and Fig. S4

*** You mention that the data generated in this study can be downloaded from Zenodo (10.5281/zenodo.10825295), but it doesn't seem to be publicly available, so I cannot review it. Please make these files publicly available at this stage.

Please also ensure that your Data Statement in the submission system accurately describes where your data can be found.

CODE POLICY

You mentioned that the code used to process the data and generate all figures is available at https://github.com/roynov01/Apicome. Please note that we cannot accept sole deposition of code in GitHub, as this could be changed after publication. However, you can archive this version of your publicly available GitHub code to Zenodo. Once you do this, it will generate a DOI number, which you will need to provide in the Data Accessibility Statement (you are welcome to also provide the GitHub access information). See the process for doing this here: https://docs.github.com/en/repositories/archiving-a-github-repository/referencing-and-citing-content

Please confirm if the data deposited in GitHub is also included in Zenodo, and please make this data publicly available. At the moment, we cannot review it.

Reviewers' comments

Rev. 1:

The authors have adequately addressed the comments.

Rev. 2:

Exceptional responses to the many concerns of the three reviewers. I have no further critique with all my concerns answered. Only question, is why Fig 4a repeated as Supplementary Figure 7A.

Rev. 3:

The authors present an improved version of the manuscript that addresses my previous points.

---

## [Editor Report · Decision Letter 3]

15 Nov 2024

Dear Dr Itzkovitz,

Thank you for the submission of your revised Short Report entitled "Intra-cellular polarization of RNAs and proteins in the human small intestinal epithelium" for publication in PLOS Biology. On behalf of my colleagues and the Academic Editor, François Schweisguth, I am delighted to let you know that we can in principle accept your manuscript for publication, provided you address any remaining formatting and reporting issues. These will be detailed in an email you should receive within 2-3 business days from our colleagues in the journal operations team; no action is required from you until then. Please note that we will not be able to formally accept your manuscript and schedule it for publication until you have completed any requested changes.

PRESS

Sincerely, 

Ines

--

Ines Alvarez-Garcia, PhD

Senior Editor

PLOS Biology
